# Predictive Value of Precision-Cut Lung Slices for the Susceptibility of Three Animal Species for SARS-CoV-2 and Validation in a Refined Hamster Model

**DOI:** 10.3390/pathogens10070824

**Published:** 2021-06-30

**Authors:** Nora M. Gerhards, Jan B. W. J. Cornelissen, Lucien J. M. van Keulen, José Harders-Westerveen, Rianka Vloet, Bregtje Smid, Stéphanie Vastenhouw, Sophie van Oort, Renate W. Hakze-van der Honing, Jose L. Gonzales, Norbert Stockhofe-Zurwieden, Rineke de Jong, Wim H. M. van der Poel, Sandra Vreman, Jeroen Kortekaas, Paul J. Wichgers Schreur, Nadia Oreshkova

**Affiliations:** 1Wageningen Bioveterinary Research, Houtribweg 39, 8221 RA Lelystad, The Netherlands; jan.cornelissen@wur.nl (J.B.W.J.C.); lucien.vankeulen@wur.nl (L.J.M.v.K.); jose.harders@wur.nl (J.H.-W.); rianka.vloet@wur.nl (R.V.); bregtje.smid@wur.nl (B.S.); stephanie.vastenhouw@wur.nl (S.V.); sophie.vanoort@wur.nl (S.v.O.); Renate.hakze@wur.nl (R.W.H.-v.d.H.); jose.gonzales@wur.nl (J.L.G.); norbert.stockhofe@wur.nl (N.S.-Z.); rineke.jong@wur.nl (R.d.J.); wim.vanderpoel@wur.nl (W.H.M.v.d.P.); sandra.vreman@wur.nl (S.V.); jeroen.kortekaas@wur.nl (J.K.); paul.wichgersschreur@wur.nl (P.J.W.S.); 2Laboratory of Virology, Wageningen University, Droevendaalsesteeg 1, 6708 PB Wageningen, The Netherlands

**Keywords:** SARS-CoV-2, hamster model, precision-cut lung slices, activity, histology scores

## Abstract

In assessing species susceptibility for severe acute respiratory syndrome coronavirus 2 (SARS-CoV-2), and in the search for an appropriate animal model, multiple research groups around the world inoculated a broad range of animal species using various SARS-CoV-2 strains, doses and administration routes. Although in silico analyses based on receptor binding and diverse in vitro cell cultures were valuable, exact prediction of species susceptibility based on these tools proved challenging. Here, we assessed whether precision-cut lung slices (PCLS) could facilitate the selection of animal models, thereby reducing animal experimentation. Pig, hamster and cat PCLS were incubated with SARS-CoV-2 and virus replication was followed over time. Virus replicated efficiently in PCLS from hamsters and cats, while no evidence of replication was obtained for pig PCLS. These data corroborate the findings of many research groups that have investigated the susceptibility of hamsters, pigs and cats towards infection with SARS-CoV-2. Our findings suggest that PCLS can be used as convenient tool for the screening of different animal species for sensitivity to newly emerged viruses. To validate our results obtained in PCLS, we employed the hamster model. Hamsters were inoculated with SARS-CoV-2 via the intranasal route. Susceptibility to infection was evaluated by body weight loss, viral loads in oropharyngeal swabs and respiratory tissues and lung pathology. The broadly used hamster model was further refined by including activity tracking of the hamsters by an activity wheel as a very robust and sensitive parameter for clinical health. In addition, to facilitate the quantification of pathology in the lungs, we devised a semi-quantitative scoring system for evaluating the degree of histological changes in the lungs. The inclusion of these additional parameters refined and enriched the hamster model, allowing for the generation of more data from a single experiment.

## 1. Introduction

In December 2019, the first cases of atypical pneumonia associated with infection by a novel coronavirus (CoV) were reported in China [1]. Sequence analysis of the new CoV revealed a close phylogenetic relationship to SARS-CoV (Severe Acute Respiratory Syndrome Coronavirus)-like bat CoV strains [2,3]. Subsequently, the new CoV was officially named SARS-CoV-2 [4]. 

In the past two decades, two other coronaviruses have emerged as novel human pathogens. The first SARS-CoV emerged in 2002 and ultimately resulted in more than 8000 confirmed cases, with an apparent 10% case–fatality ratio. Fortunately, SARS-CoV transmission was effectively controlled by appropriate international responses involving quarantine measures and it was last reported in April 2004 [5]. In 2012, another CoV emerged in Saudi Arabia, named Middle East Respiratory Syndrome (MERS)-CoV. MERS-CoV causes sporadic cases of a severe respiratory syndrome and outbreaks in healthcare settings [6]. Both viruses are suspected to originate from bats and to have acquired adaptations that allow transmission to other species [7]. The natural host of the new coronavirus, SARS-CoV-2, is yet to be identified, but the virus sequence is very similar to a SARS-CoV-like virus found in bats (bat coronavirus RaTG13) and pangolins [8,9,10]. The above-mentioned examples underscore the remarkable plasticity of coronaviruses and their ability to easily switch between hosts. Thus, different animal species may play a role in future coronavirus outbreaks and also can be used as suitable models for studying the infection, transmission and pathogenesis of the corresponding viruses. 

Infection of humans with SARS-CoV-2 causes Coronavirus Disease 19 (COVID-19). The clinical symptoms of COVID-19 can vary substantially, from mild respiratory disease to acute respiratory distress syndrome (ARDS) and death. SARS-CoV-2 spread globally within one year, resulting in a pandemic with 181.3 million confirmed cases and 3.9 million deaths (Source: John Hopkins University and Medicine website, as of 28 June 2021). As soon as it became clear that the new coronavirus was rapidly spreading and would not be easily contained, a global effort in the scientific community arose to find suitable animal models for studying the pathogenesis and transmission of SARS-CoV-2. Diverse in silico predictions of species susceptibility were published [11,12,13,14,15]. Based on in silico modeling of interactions between the receptor for the virus, angiotensin converting enzyme 2 (ACE2) and the viral receptor-binding domain (RBD), these studies aimed at identifying animal species that are susceptible to SARS-CoV-2 infection. In parallel, many labs worldwide started exploring different animal species in the search for those that best reproduce the moderate to severe human disease. A starting point for many of these studies was results obtained previously with SARS-CoV, since this virus utilizes the same receptor, ACE2. 

To tackle the initial lack of animal models, to minimize the ethical concerns of using irrelevant animal species and to reduce the number of animals used for the development of models, we evaluated the use of precision-cut lung slices (PCLS) as an ex vivo model to predict the susceptibility of different animal species to infection with SARS-CoV-2. In PCLS, the lung 3D architecture and physiology are largely preserved, and disease pathogenesis at least partly can be mimicked, rendering this system suitable for studying chronic inflammatory conditions such as asthma, chronic obstructive pulmonary disease (COPD) and idiopathic pulmonary fibrosis, but also for toxicological studies and infectious diseases (for a recent review, see [16]). Although the PCLS can be cultivated for only relatively short periods of time (up to 14 days), infection of PCLS can be reliably quantified by PCR or visualized by immunohistochemistry (IHC) and/or immunofluorescence (IF). For preparation of PCLS, animals still need to be sacrificed, but several experiments can be performed with samples collected from only one donor, and multiple variables can be tested concurrently, as opposed to in vivo experiments. Moreover, the donor animals are euthanized without any discomfort related to infection and disease. 

In this study, we evaluated the susceptibility of PCLS derived from pigs, hamsters and cats to infection with a Dutch SARS-CoV-2 isolate. Virus replication was assessed by PCR and immunofluorescence (IF). The results obtained from experiments with PCLS demonstrate that this ex vivo model can be successfully employed to predict the susceptibility of different animal species to SARS-CoV-2 and possibly other betacoronaviruses. Furthermore, we verified the susceptibility of hamsters from the same genetic background as used for the PCLS preparations. We infected adult and juvenile hamsters to look for possible age-related clinical differences, as seen in humans. Next to using well-defined parameters to establish the infection progress, we enriched and refined the hamster model by adding measurements of the individual animal activity by an activity wheel and demonstrated that daily activity counts can be employed as a clinical readout parameter of animal wellbeing post-challenge infection. Furthermore, we propose a scoring system for the quantification of lung pathology.

## 2. Results

### 2.1. Virus Isolation and Characterization

To obtain a SARS-CoV-2 isolate, nasal and oropharyngeal swab samples were collected from a Dutch person with clinical symptoms of COVID-19 during the first pandemic wave in the Netherlands (March 2020). The virus, referred to as SARS-CoV-2/human/NL/Lelystad/2020 (strain Lelystad), was isolated on Vero-E6 cells and a stock was prepared by two low-MOI passages. Infectious titers were between 5 and 5.7 log10 TCID_50_/mL. Next-generation sequencing (NGS) results confirmed the absence of adventitious agents and showed that the consensus sequences of the two passages were 100% identical and that no low abundance mutations were present in the viral spike protein. The isolate belongs to lineage B.1.22 according to the nomenclature of Rambaut et al. [17] and clusters with other European isolates corresponding to the Italian outbreak from the beginning of 2020. The genome sequence was deposited in GenBank under accession number MZ144583. 

### 2.2. SARS-CoV-2 Replicates in Cat and Hamster PCLS but Not in Pig PCLS

To assess the susceptibility of hamster, pig and cat lung tissue for SARS-CoV-2, we infected PCLS derived from the different species with SARS-CoV-2 (strain Lelystad) and assessed viral RNA loads over time (Figure 1a). Three hamster donors, two pig donors and one cat donor were used, and experiments were performed in quadruplicate per timepoint (24, 48 and 96 h post-infection). The SARS-CoV-2 total E gene PCR analyses of samples collected at several timepoints revealed no evidence of virus replication in pig lung slices, based on the observed significant decrease in viral RNA copy numbers in time (Figure 1b). A gradual increase in viral RNA signal over time was observed in hamster and cat lung tissues. Virus titers increased more rapidly in cats than in hamsters, although in both species a significant increase in RNA viral loads was detected at 96 h post-infection. The results from the total E gene PCR were fully in line with results obtained with the subgenomic PCR. The latter technique is indicative of virus replication [18].

### 2.3. SARS-CoV-2 Mainly Replicates in Alveolar Epithelium in Both Hamster and Cat PCLS

To visualize target cells with active virus replication, formalin-fixed, paraffin-embedded PCLS were subjected to immunostaining of viral antigen by immunofluorescence (IF). IF analysis confirmed the absence of viral antigen in pig PCLS and the presence of viral antigen in both hamster and cat tissue, most predominant at 3–4 days post-infection (Figure 2). In hamsters, viral antigen was detected in the alveolar epithelium and occasionally in bronchiolar epithelial cells, predominantly on day 3 post-infection. In cats, the antigen was detected in the alveolar epithelium, lung macrophages and epithelium of the bronchiole. Altogether, these results confirm the PCR data and show that SARS-CoV-2 virus replicates in pneumocytes and the bronchiolar epithelium of susceptible animal species.

### 2.4. Susceptibility of Juvenile and Adult Syrian Hamsters for SARS-CoV-2 (Strain Lelystad)

#### 2.4.1. Clinical Signs of Disease

To validate the predictability of the results obtained from experiments with PCLS and to develop an animal model based on the same SARS-CoV-2 strain and hamster breed, Syrian hamsters were intranasally inoculated with 10^4.5^ TCID_50_ of SARS-CoV-2 strain Lelystad (Figure 3a). Eighteen juvenile and eighteen adult hamsters were inoculated, anticipating on potential age-related differences in susceptibility or clinical disease. Two hamsters per age group served as uninfected controls. No mortality was observed post virus infection. All hamsters lost body weight as a result of the infection. The maximum body weight loss for individual hamsters varied between days post-infection (DPI) 3 and 7 (Figure 3b). No significant difference was observed between the peak body weight loss in adult and juvenile hamsters, although a larger variation was observed in the group of younger animals (mean body weight loss: 14.2% (SD = ±3.1) for adult and 13.4% (SD = ±6.7) for juvenile hamsters) (Figure 3c). Body weights of juvenile hamsters gradually returned to baseline levels around DPI 10. On DPI 10, adult hamsters had an average body weight of 94.4% of the baseline body weight and 97.6% at the end of the experiment (DPI 21). Of note, the juvenile hamsters were housed in activity tracking cages, while the adult hamsters were housed in regular cages without a running wheel. 

Activity was measured only in juvenile hamsters. The average number of rotations of the activity wheels before challenge was 44,283 per 24 h (Figure 3d). Activity reduction was observed in all hamsters after challenge. At DPI 2, a severe drop in activity was observed (average rotations 276 per 24 h). The activity remained low until DPI 6 (7487 average rotations per 24 h between days 3 and 6) and then gradually increased and reached levels before challenge on DPI 9. From the eight hamsters in activity cages, four had temperature transponders implanted in their abdominal cavity. No adverse effects of the implanted transponder on activity were noticed throughout the study (Figure 3d).

Core body temperature was followed by temperature transponders implanted in the abdominal cavity of four adult and four juvenile hamsters. No increase in body temperature was observed post inoculation (Figure 3e,g). A short decrease in temperature was observed in all hamsters on DPI 0, which can be attributed to the anesthesia used during the inoculation procedure, and slowing down of the metabolic rate (Figure 3g). A second drop in body temperature was observed between DPI 2 and 5, most pronounced at DPI 4 (Figure 3e,g). This body temperature reduction coincided with the substantially reduced activity in the juvenile hamsters. No differences in the temperature drop were observed between the adult and the juvenile hamsters, when accounting for the lower base temperature of the adult hamsters (Figure 3f,h). 

#### 2.4.2. Gross Pathology Findings

As expected, no gross lesions were observed in the lungs (Figure 4a) or any other organs of the two adult and the two juvenile non-infected hamsters that were euthanized before challenge. Post-inoculation, two juvenile and two adult hamsters were euthanized on DPI 2, 3, 4, 7 and 10 each, and the remaining eight juvenile and eight adult hamsters on DPI 21. Following necropsy post-inoculation, gross pathology lesions were observed in the lungs only. These lesions were characterized by poorly collapsed lungs with multifocal red to brown foci, often located around the bifurcation of the trachea (Figure 4b). Lungs were macroscopically scored as described in the Materials and Methods section. All hamsters necropsied between DPI 3 and DPI 10 showed gross pathology lesions, most prominent from DPI 4 to DPI 10 (Figure 4c). The gross pathology scoring was corroborated by digital analysis of the consolidated lung area based on photographs (Appendix A). The relative lung weight, indicative of pneumonia, followed the same trend as the gross pathology lung scores (Figure 4d). No significant differences between adult and juvenile hamsters were noted (Figure 4c,d).

#### 2.4.3. Histopathological Analysis

As expected, histopathological evaluation of the respiratory tract of non-inoculated control hamsters showed no particular findings (Figure 5a–c). Moderate to severe interstitial pneumonia (grade 2 to 4) was observed in hamsters euthanized on DPI 4, 7 and 10 post-infection (Figure 5d), characterized by the degeneration of alveolar walls, type II pneumocyte proliferation, vasculitis, influx of heterophils and hemorrhage. Histopathological lung lesions were scored semi-quantitatively, as described in the Materials and Methods section and the scores are shown in Figure 5j,k. No substantial differences were observed in the magnitude of lung pathology between the juvenile and adult hamsters (Figure 5j), although the lesions progressed and subsequently resolved somewhat faster in the juvenile hamsters (Figure 5k). On DPI 21, only mild histopathological lung lesions were present in both age groups. 

Nasal conchae showed moderate to severe histopathological changes, characterized by the degeneration and necrosis of the epithelial layer and the accumulation of mainly heterophils in the epithelial layer and submucosa with exocytosis of heterophils in the nasal cavity (purulent rhinitis), most prominent from DPI 2 until DPI 4 (Figure 5e).

In the trachea, only mild to moderate histopathological lesions were observed from DPI 1 to 7, as determined by the loss of cilia on epithelial cells and epithelial cell degeneration (Figure 5f). An increased number of inflammatory cells in the lamina propria and, to a lesser extent, in the epithelial layer was observed (Figure 5f). 

Other investigated organs (heart, liver, spleen, kidney, mesenterial lymph node, jejunum, ileum, colon, pancreas, cerebrum and cerebellum) showed no significant histopathologic changes (data not shown).

#### 2.4.4. Immunohistopathological Analysis

Presence of viral antigen (nucleoprotein (NP)) was evaluated by IHC. No viral antigen was detected in the heart, liver, kidney, jejunum, ileum, colon, pancreas or brain (cerebrum and cerebellum) following inoculation (data not shown). Viral antigen was found in the conchae, trachea and lung (also in a tracheobronchial lymph node; however, this lymph node was not consistently sampled and scored throughout the experiment). The lungs of all hamsters showed extensive staining (grade 3 or 4) from DPI 2 until DPI 4 (Figure 5g,l), and mild (grade 1) staining on DPI 7–10, present only in a limited number of hamsters. The staining was observed predominantly in alveolar epithelial cells, bronchiolar epithelial cells and, to a lesser extent, in macrophages or desquamated alveolar epithelial cells. The IHC score in juvenile hamsters was slightly higher than in the adult hamsters early post-infection, which suggests higher viral loads in the lungs of the juvenile animals. All hamsters showed positive staining in respiratory and olfactory epithelial cells in the nasal conchae on DPI 2 until 10, which was extensive (grade 3) on DPI 2 (Figure 5h) and DPI 3 and minimal on DPI 10 (grade 1). Staining of the trachea of all hamsters was only mild and limited to the epithelial cells (grade 1) on DPI 2 (Figure 5i) and DPI 3.

#### 2.4.5. Viral RNA and Infectious Virus in Swabs and Organs

In oropharyngeal swabs obtained in the first week after inoculation, high levels of SARS-CoV-2 RNA were detected (Figure 6a and Appendix A). Around DPI 10, the RNA levels decreased, and, by DPI 21, hardly any viral RNA was detected. The subgenomic RNA (sgRNA) PCR showed a similar trend. No virus could be isolated from any of the swabs by virus isolation assay.

In nasal turbinates (conchae) collected at necropsy, the viral loads followed a pattern similar to the swabs, with high levels of viral RNA and subgenomic RNA in the first week post-inoculation (Figure 6a and Appendix A). However, in contrast to the swab samples, virus could be isolated with a virus isolation assay from nasal conchae homogenates of DPI 2, 3 and 4. Interestingly, homogenates from DPI 7 were virus isolation-negative, although levels of total viral RNA were comparable to amounts on DPI 2–4 and sgRNA was also present. Total viral RNA was still detectable at later timepoints, but the levels were at least 2 log10 lower than at earlier timepoints (Appendix A). No sgRNA was detected at DPI 10 and 21 with the exception of one adult hamster at DPI 10, and no viable virus could be recovered by virus isolation assay.

In lungs, the highest amount of total viral RNA and subgenomic RNA was detected between DPI 2 and 4 (Figure 6a and Appendix A). At later timepoints, the viral loads decreased gradually, but remained detectable by PCR on total RNA in all hamsters up to DPI 21. The sgRNA was not detectable later than DPI 4. Similar to the conchae homogenates, virus was isolated only from the lung homogenates from DPI 2, 3 and 4 and not at later timepoints.

Viral loads in the trachea followed roughly the same pattern as those in the lungs but were, on average, around 2 log10 lower (Figure 6a and Appendix A). 

From the non-respiratory tissues tested, total RNA was detected in the colon, feces, duodenum and brain (Appendix A). Subgenomic RNA was occasionally detected in the colon and feces, but not in the duodenum or brain (Appendix A). No virus could be isolated from any of these tissue homogenates at any timepoint post-inoculation by a virus isolation assay (data not shown). Overall, the viral loads in non-respiratory tissues were much lower than in lungs and conchae (Table 1).

No differences in viral loads were observed between juvenile and adult hamsters in oropharyngeal swabs and in trachea homogenates. Interestingly, the conchae had higher viral RNA levels in the adult hamsters, although the difference was significant only for the subgenomic RNA. In contrast, in lungs, higher RNA loads were observed in the juvenile hamsters (Figure 6 and Appendix A) and the difference with adult hamsters was significant for both the total E-gene and the subgenomic PCR. The average differences between adult and juvenile hamsters in lungs and conchae were consistent over time but never exceeded 1 log10 RNA copies (Appendix A) and therefore did not influence the overall detectability of the viral RNA throughout the course of infection. The detection of slightly higher viral RNA loads in the lungs of the juvenile hamsters correlated with the higher IHC score (Figure 5l), indicating that the virus probably replicated slightly more efficiently in the lungs of the younger hamsters. Of note, no differences in titers of the virus recovered by virus isolation were observed. 

#### 2.4.6. Antibody Responses

Neutralizing antibodies were detected in all hamsters sacrificed after DPI 4 (Figure 6b). Remarkably, low levels of antibodies were detected in 2 out of 4 hamsters as early as DPI 4 (MN_50_ 5.74 for both hamsters). At DPI 10, antibody titers had already reached a plateau (average MN_50_ 205 for both adult and juvenile hamsters) and remained at these levels until the end of the observation period at DPI 21 (average MN_50_ 205 and 292 for adult and juvenile hamsters, respectively). No differences in antibody levels were observed between the adult and the juvenile hamsters.

## 3. Discussion

Here, we report the evaluation of PCLS as an ex vivo system that can be used to predict the susceptibility of different animal species to SARS-CoV-2. This approach can be used for other coronaviruses or other newly emerged viruses, when suitable animal models need to be established. Furthermore, we assessed the susceptibility of juvenile versus adult Syrian hamsters using the same SARS-CoV-2 strain and hamster breed as used in the PCLS experiments. 

The experiments with PCLS derived from pigs showed no evidence of SARS-CoV-2 replication despite the fact that cell lines of pig origin (swine testicle (ST) and porcine kidney (PK-15)) were previously shown to be susceptible to SARS-CoV-2 infection [19]. Our results are, however, fully in agreement with in vivo infection experiments that showed that pigs do not support SARS-CoV-2 replication [19,20,21,22] and only occasionally seroconvert [22,23]. A recent report showed that SARS-CoV-2 was recovered from the submandibular lymph node of one pig 13 days after inoculation, and viral RNA was detected in nasal washes of 2 out of 16 pigs and in the oral fluid of 1 out of 16 pigs on DPI 3 [23]. Furthermore, specific antibodies were detected in the serum of 2 out of 16 pigs and in the oral fluid of one pig. Nevertheless, these so far rare events are still in line with the overall very low susceptibility of pigs to infection with SARS-CoV-2 and with our results of PCLS infection. 

In addition to PCLS from pigs, we also incubated PCLS derived from domestic cats with SARS-CoV-2. The virus efficiently replicated in cat pulmonary tissue, probably even more efficiently compared to hamsters, although it should be noted that the data were obtained from a single donor due to the limited accessibility of cats. The cat PCLS results are in agreement with experimental findings by various groups [21,24,25,26]. Interestingly, viral antigen was detected in alveoli in cat PCLS, whereas in in vivo-infected cats, only the acinar glands seemed to support virus replication [24]. We speculate that the buffer/agarose gel of the ex vivo system might have influenced the viral attachment. Another speculation is that the alveolar walls are sensitive also in vivo, but that the limited viral replication in the large bronchi in vivo [21,24] hampers the virus infection to affect the deeper airways. Finally, PCLS are deprived of immune cell infiltrates, which might modify the dynamics of virus replication in vivo in the cat lungs.

In addition to efficient replication in cat PCLS, SARS-CoV-2 also efficiently replicated in PCLS derived from hamsters. Virus was detected in the alveolar and bronchiolar epithelium, similarly to in vivo infected hamsters, where the virus was predominantly found in the same type of cells as observed by our group and others [27,28,29].

Hamsters are widely accepted as a good model for SARS-CoV-2 infection, recapitulating mild to moderate human disease, prominent lung pathology, efficient virus replication and seroconversion. Our in vivo data obtained with the Lelystad strain of SARS-CoV-2 are in good agreement with the findings of other groups with various strains [27,28,29,30,31]. As a clinical readout, next to the frequently used body weight loss, here, we demonstrate that activity (measured by a wheel with a rotation counter) can be used as a very sensitive parameter of discomfort. Both body weight loss and activity measurements were sufficiently robust to be reliably used as endpoints of disease. Other clinical parameters seem less reliable. Clinical signs, although present, are mild and therefore clinical scoring is sensitive to subjectivity. Body temperature did not increase post-challenge. In contrast, a clear temperature decrease was observed between DPI 2 and 5. Hypothermia could be used as an additional readout parameter, measured together with activity and body weight. The lowest body temperatures measured post-inoculation coincided with the highest decrease in activity. However, since a surgical procedure is required to implant the temperature transponders into the abdominal cavity, it is questionable whether the temperature measurement adds sufficient value to justify the discomfort resulting from the surgical procedure of implantation.

Lung pathology and lung IHC were most prominent between DPI 4 and 7 and therefore a necropsy at any DPI between 4 and 7 should give sufficient information about lung pathology or lack thereof. Thus, timely necropsy should be applied to reduce animal discomfort when this does not interfere with the interpretation of the experimental results. To facilitate a more accurate evaluation of lung pathology, as suggested and described previously [32], we present a detailed scoring scheme that includes several parameters for quantification of lung pathology. Depending on the timepoint of necropsy after infection, one or more of these pathological parameters can be used.

SARS-CoV-2 was found to replicate exclusively in the respiratory organs (mainly conchae and lungs). Determination of viral loads in these organs by both RT-qPCR and virus isolation is robust and gives clear readout parameters. As an alternative sample for the detection of replication in the upper respiratory tract, we used oropharyngeal swabs. Collecting these samples is less invasive than nasal washes (nasal washes require anesthesia) and can therefore be used as a refinement. Virus isolation from respiratory organs was successful only until DPI 4. Virus could not be isolated from the swabs, but subgenomic RNA PCR was positive, suggesting the presence of replicating virus in cells collected during the swabbing process. Finally, hamsters seroconverted early after inoculation and the presence and/or boost of antibody titers may thus be used to evaluate vaccine efficacy. 

As part of this study, we also assessed the susceptibility of juvenile and adult hamsters to infection with SARS-CoV-2. Adult hamsters were used in an attempt to mimic the more severe symptoms most commonly reported in elderly humans after infection with SARS-CoV-2 [33,34]. Although the body weights of adult hamsters did not recover as quickly as those of juvenile hamsters, we did not observe differences in the lowest body weight between the two age groups as a result of challenge infection. The hypothermia in the adult hamsters post-infection seemed more pronounced than in the juvenile hamsters. However, the average baseline body temperature of the adult hamsters was also lower. When accounting for the difference in the baseline temperatures before challenge, the temperature reduction was not significantly different between adult and juvenile hamsters. In terms of pathology, the differences were limited to the progression of the lesions, which advanced and subsequently resolved slightly faster in juvenile hamsters. Our data also suggest that the virus may replicate slightly better in the conchae of adult hamsters and in the lungs of juvenile hamsters. It has to be noted that the statistical analysis was performed on a very limited number of samples per timepoint (N = 2 for DPI 0–10). Therefore, generalization of these observations for age-related differences should be performed with caution. Our main conclusion is that the differences between adult and juvenile hamsters are not substantial and that there is no advantage of using adult over juvenile animals in this model for SARS-CoV-2 infection. No differences between young and older hamsters have been reported by others either [29,35]. 

In conclusion, with this work, we demonstrate that PCLS can be used as a convenient substitute for the testing of animal susceptibility towards SARS-CoV-2. This approach can also be applied for other newly emerged viruses, for which good animal models are lacking. Multiple PCLS from different species can be prepared and tested before animals are used in subsequent experiments to establish animal models. By using this method, a relatively fast screening can be performed with the use of only a few animals, before experiments with many animals are undertaken. Furthermore, we describe a refined hamster model, to which we added activity tracking as a valuable clinical parameter, and a histopathological scoring system for a standardized semi-quantitative evaluation of the lung pathology.

## 4. Materials and Methods

### 4.1. Animals

For both PCLS and in vivo inoculations, Syrian hamsters (*Mesocricetus auratus*; female), strain RjHan:AURA, were obtained from Janvier (Le Genest-Saint-Isle, France). They had an SPF (Specific Pathogen-Free) health status and were either 7 weeks (juvenile) or 8 to 12 months of age (adult) at arrival. To obtain PCLS samples from pigs, conventional pigs (*Sus scrofa domesticus*; male), TOPIGS Norsvin 70, were purchased from a high health status farm (Van Beek; Lelystad, The Netherlands). Upon arrival, the pigs were 4–5 months of age. To obtain PCLS samples from cats, domestic shorthair cats (*Felis catus*; male and female) were purchased from Marshall Bioresources (North Rose, NY, USA). Cats were raised as SPF cats, vaccinated against Rabies and Feline Rhinotracheitis-Calici-Panleukopenia, and were 4 months old upon arrival.

### 4.2. Virus and Cells

A SARS-CoV-2 isolate, referred to as SARS-CoV-2/human/NL/Lelystad/2020, was used in this study. The virus stock was prepared by two low-MOI passages (p1 and p2) in Vero E6 cells (ATCC^®^ CRL-1586™; Manassas, VA, USA) after initial isolation from an oropharyngeal swab from a human subject. Cells were maintained on MEM (Gibco, RefNo 21090; Thermo Fischer Scientific; Waltham, MA, USA), supplemented with 5% FCS, 1% antibiotic/antimycotic (Gibco; Thermo Fischer Scientific; Waltham, MA, USA), 1% L-glutamine, 1% Minimal Essential Medium Non-Essential Amino Acids (MEM-NEAA) (all from Gibco; Thermo Fischer Scientific; Waltham, MA, USA). This medium is referred to as complete medium throughout the manuscript and was also used for cell culturing and virus propagation.

### 4.3. Next-Generation Sequencing (NGS)

Virus stock material from both passages (p1 and p2) was treated with benzonase (Sigma; Saint Louis, MO, USA) to remove host RNA/DNA (2 mL virus stock was incubated with 1 µL benzonase (250 U) and 2 µL 2 mM MgCl2 (Merck; Kenilworth, NJ, USA) for 4 h at 37 °C). RNA was isolated using the NucliSens easyMAG automated RNA isolation robot and NucliSENS easyMAG kit (Biomerieux; Marcy-l’Étoile, France), with 500 µL input material and 50 µL elution material. The obtained four elution fractions were pooled and concentrated using the Zymoclean RNA Clean&Concentrator kit (Zymo Research, RefNo R1013; Irvine, CA, USA). The cleaning/concentration was repeated twice to remove excessive content of small RNAs. The prepared RNA was subsequently analyzed with next-generation sequencing on an Illumina platform (MySeq System, Illumina; San Diego, CA, USA), using both an RNA-seq technique (RNA preparation with KAPA RNA HyperPrep Kit, Roche, RefNo 08098107702; Basel, Switzerland) and preparation of a cDNA library (KAPA HyperPlus Kit, Roche, RefNo 07962428001). Full sequences were obtained from both methods. 

### 4.4. Hamster Experiment

Twenty juvenile (7 weeks old) and twenty adult (8–12 months old) Syrian hamsters were housed in individual cages and chipped subcutaneously upon arrival for identification. After an acclimatization period of 7 days, four juvenile and four adult hamsters were randomly selected and were implanted with a temperature transponder in the abdominal cavity (Anipill^®^, Hérouville Saint-Clair, France). The temperature probes were set to measure the abdominal temperature every 15 min throughout the whole experiment. Seven days post-surgery, the four juvenile hamsters with the temperature transponders and another four juvenile hamsters without transponders were individually housed in cages with activity tracking wheels (Tecniplast, Buguggiate, Italy). The individual activity of the hamsters was measured by automatized counting of the wheel rotations when hamsters ran in the wheels, with four counts equaling one full rotation (perimeter approximately 97 cm). The counters were read out daily, approximately at the same time of the day (±2 h), and set back to zero. Activity was monitored from DPI −4 (days post-infection) until the end of the study (DPI 21). Body weights of the hamsters were measured daily from DPI −7 until the end of the study. Before inoculation, two juvenile and two adult hamsters were necropsied to collect uninfected tissues for comparison. All other hamsters (N = 36) were exposed to SARS-CoV-2 via intranasal inoculation with 100 µL of undiluted virus stock (dose of 10^4.5^ TCID_50_), which is the highest virus concentration that could be achieved with our virus stock. Virus was applied in a volume of 50 µL per nostril (100 µL in total) synchronous to the hamster’s breathing rhythm under anesthesia with 0.15 mg/kg medetomidine (Sedastart, ASTfarma; Oudewater, The Netherlands) and 100 mg/kg ketamine (Narketan, Vetoquinol; Breda, The Netherlands), which was antagonized by atipamezole (Sedastop, ASTfarma; Oudewater, The Netherlands). At 2, 3, 4, 7 and 10 days post-infection, two juvenile and two adult hamsters were euthanized to collect organs for pathohistological and virological analysis and blood for detection of antibodies. The remaining N = 16 hamsters (8 juvenile and 8 adult) were followed up to day 21, when they were euthanized and organs and blood was collected. Hamsters were monitored daily for general health and body weight loss. Details of the experimental scheme are shown in Figure 3a. Humane endpoints were defined as follows: four or more infection-related clinical signs (such as ruffled fur, nasal or ocular discharge, coughing, sneezing, reduced activity while being handled, curled up position, neurological signs, impaired breathing (abdominal breathing or increased frequency) and 20% or more body weight loss compared to the day of infection); lethargy for more than 24 h; neurological seizures; severe respiratory distress.

### 4.5. Preparation of Precise-Cut Lung Slices (PCLS)

Donor animals for PCLS were euthanized by deep general anesthesia (hamsters and cats: medetomidine and ketamine, pigs: tiletamine, zolazepam, xylazine) followed by exsanguination. The lungs were carefully removed and either filled completely (hamster, cat) via the bronchus with 37 °C 1.5% agarose in RPMI 1640 (type VII-A low gelling temperature, Sigma) or a right cranial lung lobe (pig) was used. After agarose solidification on ice (10 min), the tissue was sliced into cubes of around 1 cm^3^. The tissue cubes containing 1.5% agarose were subsequently embedded in 4% agarose by placing them in a 10 mL syringe with 3 mL of unsolidified 4% agarose (Figure 1a). After filling of the syringe with additional agarose, solidification was continued for 10 min at 4 °C. The embedded lung tissue was subsequently cut into slices of 350 µm using a VT1000S vibratome (Leica, Amsterdam, Netherlands) set to a frequency of 80 Hz, speed of 2.5 mm/s, angle of 1.5 degrees and with a cycle speed of 60 slices/min. Each slice was placed in one well of a 24-well plate, pre-filled with 1 ml of RPMI 1640 medium (Gibco; Thermo Fischer Scientific; Waltham, MA, USA) supplemented with 5% FBS and 1% antibiotic/antimycotic solution (Gibco; Thermo Fischer Scientific; Waltham, MA, USA). After 2–4 h incubation at 37 °C and 5% CO_2_, the slices were incubated with 100 µL SARS-COV-2 (5.5 log10 TCID_50_/mL). Following overnight incubation, slices were washed three times with PBS (to remove extracellular virions), and at 24, 48 and 96 h post-infection, three slices per timepoint were placed in Trizol and frozen at ≤−70 °C until RNA isolation. In addition to the 350 µm PCLS, thicker slices (600 µm) were also incubated with SARS-CoV-2 (100 µL; 5.5 Log10 TCID_50_/mL) for histology (hematoxylin and eosin stain) and IHC analysis following routine formalin fixation and paraffin embedding as described below.

### 4.6. Pathological and Histological Evaluation of Tissues

Upon necropsy, all major organs were examined macroscopically. Lungs were weighed and scored by a board-certified veterinary pathologist. A gross pathology score was assigned after evaluating the dorsal and ventral aspect of each lung (Table 2). For histopathology, the respiratory tract (lower half of trachea, left lung lobe inflated with 10% neutral-buffered formalin, left nasal conchae), the gastro-intestinal tract, heart, spleen, liver, kidney and brain (left sagittal section of brainstem, cerebrum, cerebellum) were fixed in 10% neutral-buffered formalin (Klinipath BV; VWR; Radnor, PA, USA) for 1 week and embedded in paraffin. Sections of 3 μm were stained with hematoxylin and eosin (H&E) for histologic analysis. All organs were evaluated and lungs were semi-quantitatively scored for extent of lung pathology and severity of different histopathological parameters: alveoli (degeneration/necrosis of alveolar wall, type II pneumocyte proliferation, presence of inflammatory cells), inflammation of bronchi/bronchiole, inflammation of blood vessels and the presence of edema (Table 2).

### 4.7. Immunohistochemistry (IHC) and Immunofluorescence (IF)

SARS-CoV-2 antigen was detected by immunohistochemistry in 10% formalin-fixed and paraffin-embedded tissue (left lung, left nasal conchae and trachea). Briefly, heat-induced epitope retrieval (HIER) method was used to prepare slides for IHC stain. After routinely dewaxing and inhibition of endogenous peroxidase (methanol/H_2_O_2_), the sections were heated for 15 min at 100 °C (Pascal, Dako, pressure cooker), in 10 mmol citrate buffer pH 6.0 (Dako S1699; Agilent; Santa Clara, CA, USA). Subsequently, the slides were blocked with 10% normal goat serum (Dako; Agilent; Santa Clara, CA, USA) and, after this, the primary polyclonal antibody rabbit anti-SARS-CoV NucleoProtein (Sino Biological, 40163-T62; Beijing, China) was added at a dilution of 1:2500 for 45 min at room temperature (RT). As secondary antibody, rabbit Envision peroxidase polymer (Dako, K4003; Agilent; Santa Clara, CA, USA) was used for 30 min at RT. The reaction was revealed with 3.3’-diaminobenzidine (DAB) (Dako K3468; Agilent; Santa Clara, CA, USA) as substrate. Slides were counterstained with hematoxylin. For immunofluorescence, sections were incubated with Alexa Fluor™ tyramide reagent (Invitrogen, Carlsbad, CA, USA) and mounted in antifading mounting medium containing DAPI (Vector laboratories, Peterborough, UK). Sections were photographed with an Olympus BX51 (fluorescence) microscope (Olympus corporation, Shinjuku, Tokyo, Japan) equipped with a high-resolution digital camera. Monochromatic digital photographs for immunofluorescence were false-colored using CellSense^®^ software (Olympus corporation, Shinjuku, Tokyo, Japan). Lungs were scored semi-quantitatively for antigen expression in the lungs, as shown in Table 3.

### 4.8. Organ Suspensions and Oropharyngeal Swabs

The following organs were collected for evaluation of viral loads: respiratory tract (upper half of trachea, right lung lobes, right nasal conchae), gastro-intestinal tract (duodenum, colon spiral), brain (right part of the brainstem, cerebrum, cerebellum after a sagittal section). The organs were kept on melting ice during necropsy and subsequently stored at ≤−70 °C until further processing. To prepare organ homogenates, each organ was weighed individually and then added to 6 mL of MEM, supplemented with 1% antibiotic/antimycotic solution (Gibco; Thermo Fischer Scientific; Waltham, MA, USA). When suspensions of intestines or feces were prepared, 100 µg/mL gentamycin was added to the medium. The organs/feces were ground for 50 s at 6000 rpm using Ultra Turrax tubes and an Ultra Turrax Tube Drive (IKA; Staufen, Germany). Lungs were ground for a shorter period of time (30 s) because of their less compact consistency compared to other organs. All organ suspensions were cleared by centrifuging for 15 min at 3400× *g* at 4 °C. Subsequently, cleared organ homogenates were either directly suspended in Trizol-LS (Sigma; Saint Louis, MO, USA) at a ratio of 1:3 (one part suspension and 3 parts Trizol) and stored at ≤−15 °C until RNA isolation or directly frozen at ≤−70 °C for virus isolation.

Oropharyngeal swabs (MW100 DryswabTM Fine Tip, sterile, MWE, Essex, UK) were obtained from the hamsters and directly submerged in 2 mL cell culture medium. During necropsy, the tubes with swabs were kept on melting ice until further processing. In the lab, the tubes were vigorously vortexed for 30 s on a vortex (Labdancer, VWR) and then centrifuged for 5 min at 1500× *g* and 4 °C in a precooled centrifuge. Aliquots for RNA or virus isolation were prepared as described above for the organ suspensions.

### 4.9. RNA Extraction and PCR

#### 4.9.1. RNA Extraction

From Trizol-LS-lysed samples, total RNA was extracted with the Direct-zol™ RNA MiniPrep kit (Zymo Research, RefNo R1013; Irvine, CA, USA) according to the manufacturer’s instructions, without DNase treatment. RNA was stored at ≤−70 °C until used for PCR. 

#### 4.9.2. PCR on E Gene RNA (Total E Gene PCR) 

To detect the viral genomic RNA, we used the method described by Corman et al. [36] with the E_Sarbeco primer/probe set. Of note, this PCR detects both genomic viral RNA and subgenomic RNA of the E gene, generated during virus replication. A one-step reverse-transcriptase quantitative PCR (RT-qPCR) protocol was used with the TaqMan^®^ Fast Virus 1-Step Master Mix (Applied Biosystems; Foster City, CA, USA), 500 mM of the forward and reverse primers, 250 nM probe and 5 µL RNA template. The RT-qPCR was performed on a LightCycler480 platform with the following cycling conditions: 55 °C 10 min, 95 °C for 2 min, 95 °C for 15 s and 58 °C for 30 s for 40 cycles. Primers were synthesized by Eurogentec (Luik, Belgium). All PCR results were expressed as RNA copy number per ng RNA (PCLS), per swab or per gram tissue (organs). Quantifications were performed based on a standard curve that was included in each PCR run. The standard curve was prepared by 10-fold serial dilutions of RNA isolated from a virus stock (SARS-CoV-2/human/NED/Lelystad/2020) and calibrated against an RNA standard obtained from the European Virus Archives (EVAg; Marseille, France) [36].

#### 4.9.3. PCR on Subgenomic E-Gene RNA (Subgenomic PCR) 

For detection of subgenomic (sg) RNA produced by the virus during genome replication, the procedure described by Wölfel et al. [18] was used, with the following modifications: the TaqMan^®^ Fast Virus 1-Step Master Mix (Applied Biosystems; Foster City, CA; USA) was utilized and the cycling conditions were the same as described above for the total viral RNA detection. Primers were synthesized by Eurogentec (Luik, Belgium). For the sgRNA PCR, a standard curve was prepared from serial 10-fold dilutions of synthetic RNA with a sequence identical to the PCR amplicon. The concentration of the synthetic RNA was determined photometrically. A standard curve was included in each PCR run.

### 4.10. Virus Isolation and Quantification

Clarified organ suspensions were serially diluted (1:10 initial and 4-fold dilutions) in complete cell culture medium. From each suspension, three independent dilutions were prepared and 50 µL of each dilution was added to wells of 96-well plates containing 15,000 Vero E6 cells per well in 100 µL. After 1.5 h incubation at 37 °C and 5% CO_2_, the supernatants from the first two serial dilutions (1:10 and 1:40) were replaced with fresh culture medium. The monolayers were subsequently incubated for 6 days and then fixed with 4% formaldehyde (VWR; Radnor, PA, USA) (15 min incubation), followed by fixation and permeabilization with ice-cold 100% methanol for 10 min and subsequent washing with PBS. For titer determination, the plates were stained with an immunoperoxidase monolayer assay (IPMA; see below) and scored for positive (stained) wells under the microscope. Each well where at least one focus of infection was found was scored as positive. Titers were calculated using the Spearman–Kärber algorithm and were expressed as TCID_50_/mL.

### 4.11. Immuno-Peroxidase Monolayer Assay (IPMA)

Fixed cell monolayers were treated with 1% Triton X-100 (MP Biomedicals LLC; Irvine, CA, USA) solution for 10 min (RT), washed with PBS (media preparation at Wageningen Bioveterinary Research (WBVR, Lelystad, The Netherlands)) containing 0.5% Tween-80 (Sigma; Saint Louis, MO, USA) and blocked for 30 min with PBS supplemented with 5% horse serum (Sigma; Saint Louis, MO, USA) at RT. Next, the blocking solution was discarded, and the monolayers were incubated with a rabbit antiserum for 1 h at 37 °C. This rabbit antiserum (rabbit-anti-SARS-CoV-2-S1-2ST (619F)) was raised against the S1A subdomain of the SARS-CoV-2 spike protein (residues 1-294), fused to a triple Strep-Tag and produced in HEK293T cells (Davids Biotechnologie GmbH). After washing the plates three times with PBS-Tween, the monolayers were incubated with goat-anti-rabbit-HRP (Dako; Agilent; Santa Clara, CA, USA) for 1 h at 37 °C. After another three washes with PBS-Tween solution, a freshly prepared AEC (3-Amino-9-ethylcarbazole) substrate solution (19 mL substrate buffer (0.05 M NaAc buffer, pH adjusted to 5.0 using 0.05 M HAc) + 1 mL 4 mg/mL AEC (Sigma; Saint Louis, MO, USA) stock solution in DMSO (MP Biomedicals; Santa Ana, CA, USA) + 50 µL 3% H_2_O_2_ (Merck; Kenilworth, NJ, USA)) was added and the monolayers were incubated until clear red-brown color developed (usually within 30–40 min). Plates were evaluated under a standard light microscope.

### 4.12. Detection of Neutralizing Antibodies 

Upon necropsy of hamsters, blood was collected from the aorta in VACUETTE^®^ CAT Serum Clot Activator tubes (Greiner Bio One; Krensmünster, Austria). Blood was allowed to clot for 1 h at RT and separated by subsequent centrifugation for 10 min at 1250× *g* at RT. Resultant serum aliquots were stored at ≤−15 °C before heat inactivation for 2 h at 56 °C before analysis. 

Virus neutralization tests (VNT) were performed in 96-well plates by serially diluting serum samples in complete medium (initial 1:10 and then 3-fold serial dilutions; 50 µL per well). Each sample was diluted in two independent replicates. Subsequently, diluted sera were combined with 50 µL of SARS-CoV-2 in complete medium, at a dose of ~100 TCID_50_. After incubating for 1.5 h at RT, 15,000 Vero E6 cells/well in 50 µL complete medium were added to each well. Plates were incubated for 4 days at 37 °C and 5% CO_2_ before fixation with 4% formaldehyde (Boom, Meppel, the Netherlands) (15 min incubation), followed by fixation and permeabilization with ice-cold 100% methanol (10 min). After washing with PBS, the plates were stained with IPMA (see above). The titer of each duplicate was determined as the reciprocal value of the last dilution that showed ≥50% neutralization, as assessed visually under a standard light microscope. The titer of each sample was calculated as the average of the duplicate titers after log transformation and expressed as virus microneutralization titer 50 (MN_50_).

### 4.13. Statistical Analysis

Virus growth in PCLS over time (Figure 1b,c) was assessed per species, by fitting a linear regression model, where the mean RNA copy numbers (total E-gene PCR or subgenomic PCR) at 48 and 96 h were compared to the mean copy number at 24 h.

Comparison between the greatest body weight loss, as well as the temperature drop at DPI 4 in adult and juvenile hamsters was performed with a *t*-test after checking the data’s normality using the D’Agostino–Pearson omnibus K2 test.

To compare differences in pathological scores or relative lung weights between adult and juvenile hamsters (Figure 4c,d and Figure 5j–l), linear or Poisson regression models were used. In these models, the response was the pathological score/relative lung weights and the explanatory variables were the age of the hamsters, time post-infection and their interaction. To account for non-linearity in time post-infection, basic spline terms were used. The model (linear or Poisson) that best fit the data (based on Akaike’s information criterion) was chosen for each particular dataset. The model variable significance was assessed using the ANOVA test.

To compare differences in viral load dynamics in respiratory organs and oropharyngeal swabs between adult and juvenile hamsters in time post-infection (Figure 6a), a multivariate linear regression model was fit, where RNA copies or VI units were the response variable and days post-infection and age of the hamsters (juvenile/adult) were the explanatory variables. An identical approach was used for the neutralizing antibody titers (Figure 6b), but for the analysis, the titer values were log-transformed by taking the natural log of each titer. To account for non-linearity in the relationship between time and virus copies/antibody titers, natural splines were used on the variable days post-infection. The model variable significance was assessed using the ANOVA test.

Statistical analysis was performed using the statistical software R version 4.0.2 [37] to fit the linear and Poisson regression models. For the *t*-tests and all figures, GraphPad Prism version 8.3.0 was used (GraphPad, San Diego, CA, USA).

## Figures and Tables

**Figure 1 pathogens-10-00824-f001:**
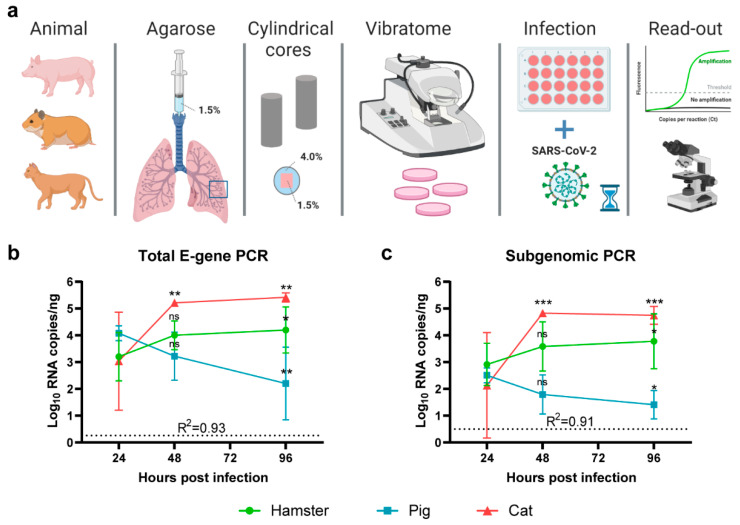
Susceptibility of PCLS to SARS-CoV-2. (**a**) Schematic presentation of the experimental setup. Briefly, pig, hamster and cat lung tissues were injected with low-melting-temperature agarose (1.5%) and embedded in cylindrical agarose cores (agarose 4%). PCLS were subsequently obtained using a vibratome and directly placed in culture medium in 24-well plates. Following virus infection, total RNA samples were obtained and assessed for SARS-CoV-2 RNA, or tissue samples were fixed in formalin and embedded in paraffin for antigen detection by IF. Three hamsters, two pigs and one cat were used as lung donors and experiments were performed in quadruplicate. (**b**,**c**) SARS-CoV-2-specific RT-qPCR analysis of PCLS, detecting (**b**) total viral E-gene RNA or (**c**) subgenomic E-gene RNA. Viral genome copies were normalized per ng RNA isolated from each slice. For hamsters and pigs, three and two independent experiments were performed, respectively, using a separate donor per experiment, while, for the cat, due to limited sample availability, one experiment was performed with one donor. Symbols represent means and error bars represent 95% confidence intervals (CI). Horizontal dotted lines show the detection limit of the respective PCR per reaction. For statistical analysis, linear regression models were fitted, where the mean RNA copy numbers at 48 and 96 h were compared to the mean copy number at 24 h. Comparisons were made within each species PCLS only. The adjusted R-squared value for the models is shown in the respective graphs. Differences with *p* values ≤ 0.05 were considered significant. Significance code: *—*p* ≤ 0.05; **—*p* ≤ 0.01; ***—*p* ≤ 0.001; ns—not significant.

**Figure 2 pathogens-10-00824-f002:**
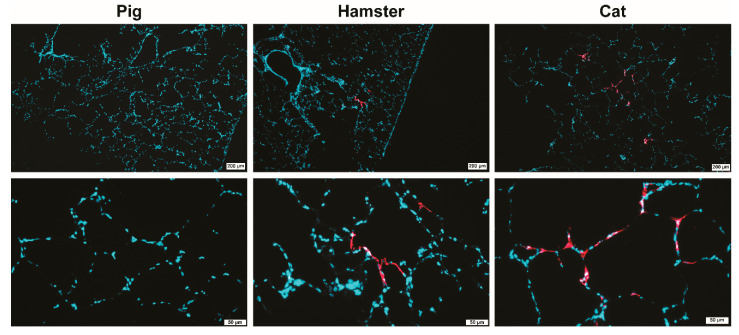
Analysis of SARS-CoV-2 infected PCLS by immunofluorescence staining for SARS-CoV-2 nucleoprotein (NP). PCLS from pig (left panel), hamster (middle panel) and cat (right panel) at 4 days post-infection are shown. SARS-CoV-2 NP is stained in red, nuclei in blue (DAPI). Upper panel—low-magnification pictures (bars 200 µm), lower panel—higher magnifications (bars 50 µm).

**Figure 3 pathogens-10-00824-f003:**
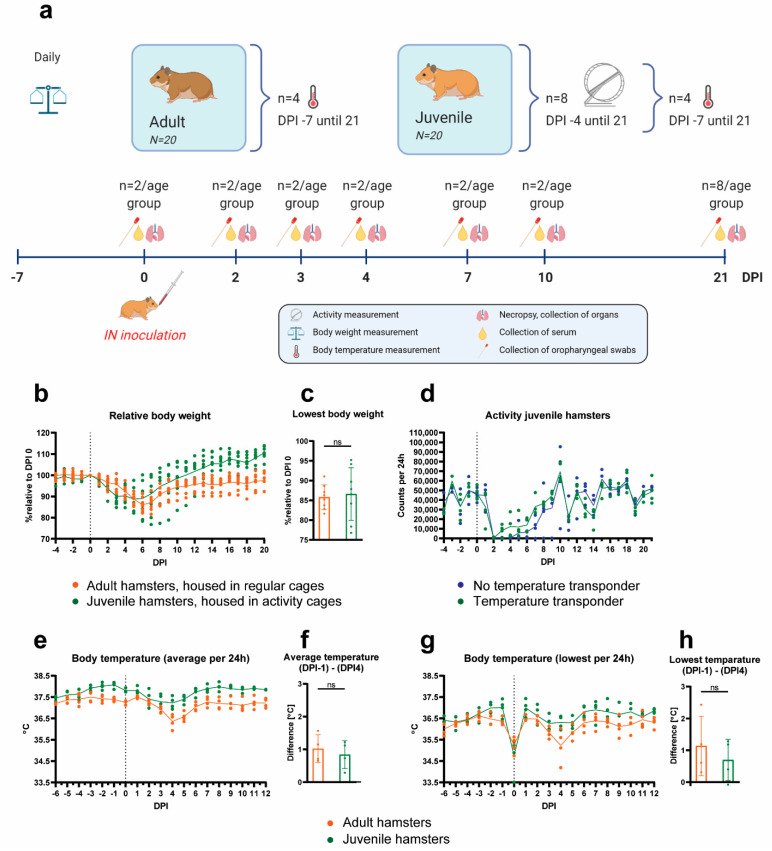
Experimental setup and clinical signs of SARS-CoV-2-infected Syrian hamsters. (**a**) Schematic presentation of the experimental setup. In brief, two juvenile and two adult hamsters were euthanized before virus inoculation on day 0 post infection (DPI 0) to collect negative control samples. Eighteen juvenile and eighteen adult hamsters were inoculated with SARS-CoV-2 via the intranasal (IN) route. Two animals of each group were euthanized on DPI 2, 3, 4, 7 and 10. The remaining eight hamsters per group were euthanized on DPI 21, of which four juvenile and four adult hamsters had an implanted abdominal temperature probe. The eight juvenile hamsters that were euthanized on DPI 21 had access to a running wheel. (**b**) Relative body weights compared to the day of virus inoculation (N = 8 per age group). (**c**) Lowest body weight that each hamster reached throughout the whole observation period (N = 8 per age group), expressed as percentage relative to the day of virus inoculation. (**d**) Individual activity counts of juvenile hamsters with (N = 4) or without (N = 4) temperature transponder. (**e**–**h**) Core body temperatures measured by temperature transponders implanted in the abdominal cavity (N = 4 per age group). Temperatures were recorded in 15 min intervals. (**e**) Average temperatures per animal per 24 h. (**f**) Average temperature difference, calculated between the most pronounced temperature drop at DPI 4 and the baseline at DPI −1. (**g**) Lowest abdominal body temperatures, measured per animal per 24 h. (**h**) Lowest temperature difference, calculated between the most pronounced temperature drop at DPI 4 and the baseline at DPI −1. (**b**,**d**,**e**,**g**) Symbols represent individual values and lines represent averages. The vertical dotted lines indicate the day of inoculation. (**c**,**f**,**h**) Bars represent averages and the error bars show standard deviation. Statistical analysis was performed with a *t*-test. Differences with *p* values ≤ 0.05 were considered significant; ns—not significant.

**Figure 4 pathogens-10-00824-f004:**
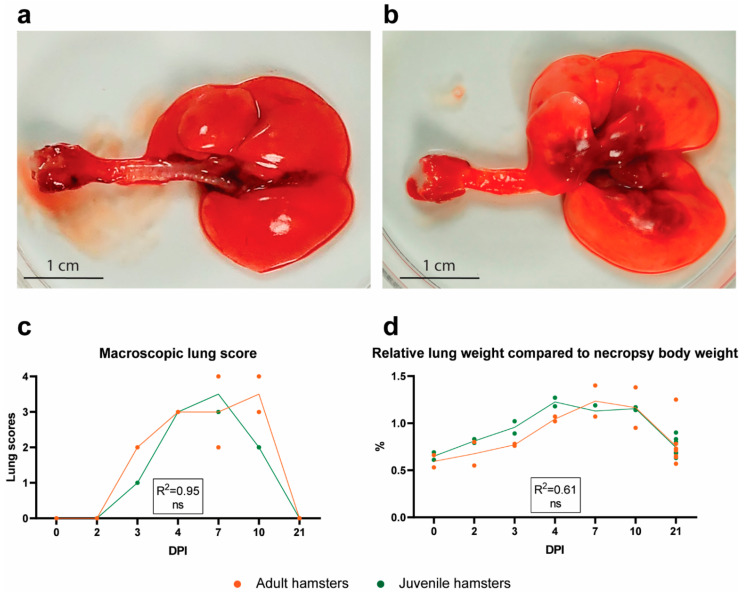
Gross lung pathology following SARS-CoV-2 infection. Dorsal view of (**a**) a non-infected lung (DPI 0) and (**b**) a SARS-CoV-2-affected lung (DPI 4). Note the prominent dark red foci around the bifurcation in the affected lung. (**c**) Macroscopic lung score (scores from 0—no changes to 4—whole lung affected). (**d**) Relative lung weights, expressed as percentage of the body weight at the day of necropsy. (**c**,**d**) Symbols represent individual values and lines represent averages of N = 2 and N = 8 hamsters per group on days post-infection (DPI) 0-10 and DPI 21, respectively. For statistical analysis, a Poisson (**c**) or a linear (**d**) regression model was used to compare differences between adult and juvenile hamsters over time. The Nagelkerke (**c**) or adjusted (**d**) R-squared value for the regression models is shown in the respective graphs. Differences with *p* values ≤ 0.05 were considered significant; ns—not significant.

**Figure 5 pathogens-10-00824-f005:**
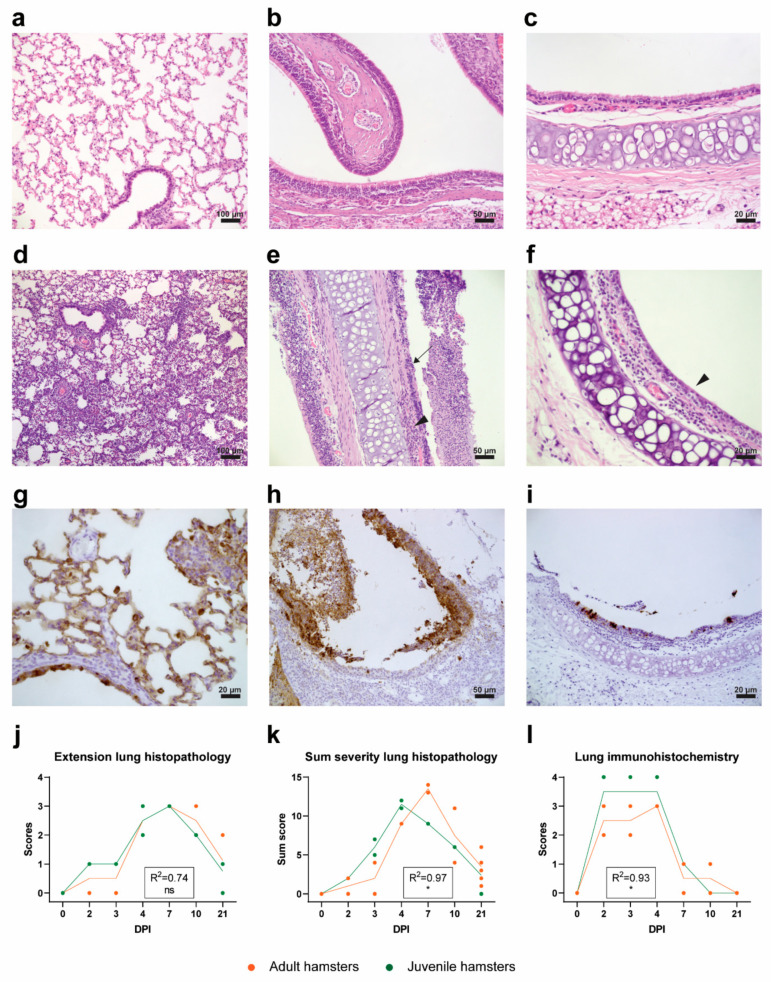
Histopathology and IHC. (**a**–**f**) H&E staining of respiratory organs. (**a**) Unaffected lung tissue of a negative control animal. (**b**) Unaffected nasal conchae of a negative control animal. (**c**) Unaffected trachea section of a negative control animal. (**d**) Severe interstitial pneumonia of a lung on DPI 4. (**e**) Severe inflammation of nasal conchae. Note the degenerated epithelial layer (arrow) and the mononuclear cell infiltration (arrowhead). (**f**) Trachea with loss of ciliated epithelium and influx of inflammatory cells in submucosa (arrowhead). (**g**–**i**) IHC staining against the nucleoprotein of SARS-CoV-2. (**g**) Staining of alveolar and bronchiolar epithelial cells (DPI 2). (**h**) Staining of nasal respiratory epithelium and desquamated epithelial cells (DPI 2). (**i**) Staining of tracheal epithelial cells (DPI 2). (**j**) Extent of lung histopathology in juvenile and adult hamsters graded from 0 (no changes) to 4 (≥70% of left lung surface is affected). (**k**) Cumulative severity of lung histopathology in juvenile and adult hamsters. (**l**) SARS-CoV-2 viral nucleoprotein expression in juvenile and adult lung tissue (IHC) graded from 0 (no staining) to 4 (extensive staining) throughout the whole tissue. (**j**–**l**) Symbols represent individual values and lines represent averages of N = 2 and N = 8 hamsters per group on days post-infection (DPI) 0-10 and DPI 21, respectively. For statistical analysis, a Poisson (**k**) or a linear (**j**,**l**) regression model was used to compare differences in the progression of histopathological lesions/scores in time between adult and juvenile hamsters in time. The Nagelkerke (**k**) or adjusted (**j**,**l**) R-squared value for the regression models is shown in the respective graphs. Differences with *p* values ≤ 0.05 were considered significant. Significance code: *—*p* ≤ 0.05; ns—not significant.

**Figure 6 pathogens-10-00824-f006:**
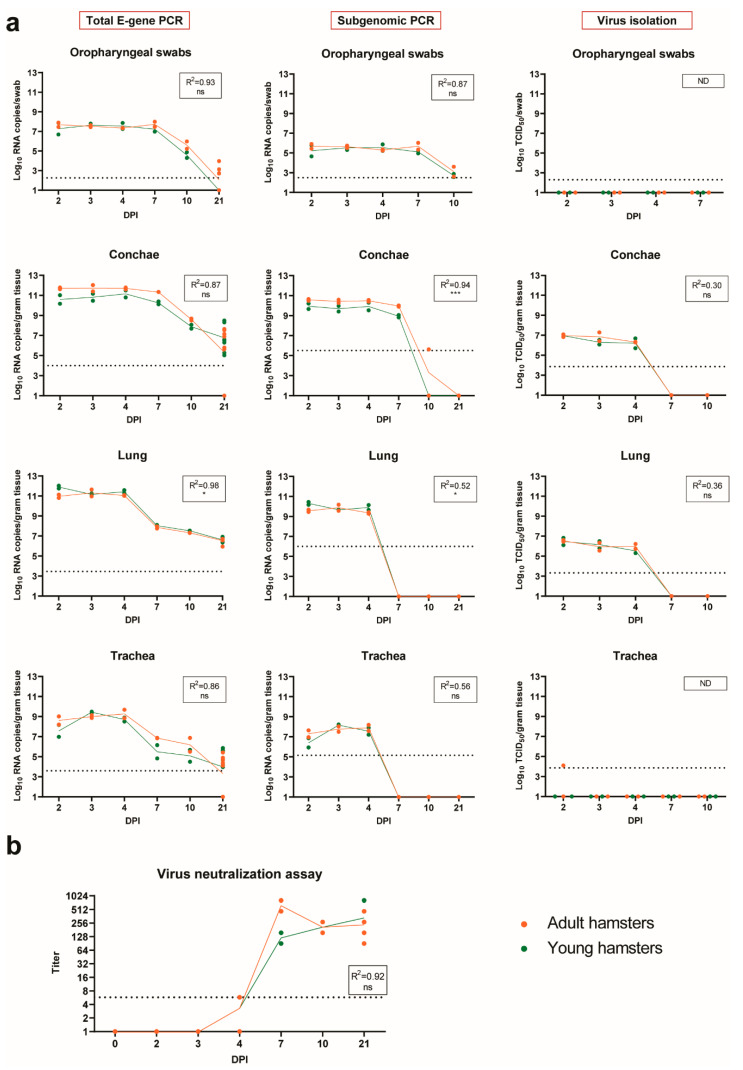
SARS-CoV-2 viral loads in oropharyngeal swabs and respiratory tissues and detection of neutralizing antibodies. (**a**) Total viral RNA (left panels), sgRNA (middle panels) and virus titers (right panels) in oropharyngeal swabs, trachea, nasal turbinates (conchae) and lung, measured in N = 2 hamsters per age group on days post-infection (DPI) 0–10, and N = 8 on DPI 21. RNA quantities were determined by RT-qPCR and are expressed as log10 RNA copies per swab or per gram of tissue. Virus titers were determined by virus isolation. Samples displayed with a value of “1” had undetectable virus or viral RNA content. (**b**) Neutralizing antibody titers as determined by virus neutralization tests and expressed as 50% neutralization titer (MN_50_). Samples displayed with a value of “1” had undetectable antibody titer. (**a**,**b**) Each symbol represents a value of an individual animal and lines show averages. Horizontal dotted line shows the detection limit of the test. For statistical analysis, linear regression model was used to compare differences in the viral load dynamics between adult and juvenile hamsters in time. The adjusted R-squared value for the regression models is shown in the respective graphs. Differences with *p* values ≤ 0.05 were considered significant. Significance code: *—*p* ≤ 0.05; ***—*p* ≤ 0.001; ns—not significant; ND—not done.

**Table 1 pathogens-10-00824-t001:** Total and subgenomic viral RNA copy number (log10) detected in different hamster tissues. The numbers are averages from all hamsters necropsied on DPI 2, 3 and 4 (peak of virus replication). Numbers in brackets show the standard deviation.

	Swabs	Trachea	Lung	Conchae	Brain	Duodenum	Colon	Feces
total RNA	7.5 (0.34)	8.8 (0.73)	11.3 (0.37)	11.3 (0.57)	6.0 (0.91)	7.4 (0.37)	7.2 (0.73)	7.6 (0.51)
sgRNA	5.5 (0.36)	7.5 (0.68)	9.8 (0.36)	10.2 (0.43)	-	-	5.0 * (1.02)	5.5 ** (0.56)

* 3 out of 12 hamsters were found positive. The average includes only the positive values. ** 9 out of 12 hamsters were found positive. The average includes only the positive values.

**Table 2 pathogens-10-00824-t002:** Scoring system used to grade lesions in H&E-stained lung tissue following infection with SARS-CoV-2.

Parameter	Score	Extent of Lesions
**General scores**		
Gross pathology	0	No macroscopical changes
	1	Focal discoloration of the lung in <15% of lung surface
	2	Multifocal discoloration of approximately 15–40% of lung surface
	3	Multifocal discoloration affecting approximately 40–70% of lung surface
	4	Whole lung affected in >70% of lung surface
Extent lung histopathology (general score, percentage) (H&E)	0	No changes
	1	Focal to multifocal (<15% of tissue)
	2	Multifocal (15–40% of tissue)
	3	Multifocal to coalescing discoloration (40–70% of tissue)
	4	Diffuse (>70% of tissue)
**Individual histological parameters (severity)**		
Alveoli (thickening alveolar wall, type II pneumocyte proliferation, inflammatory cells alveoli)/ interstitial pneumonia	0	No changes
	1	Focal to multifocal (<20% of tissue)
	2	Multifocal (20–50% of tissue)
	3	Multifocal to coalescing (>50% of tissue)
Bronchi and bronchioli	0	No changes
	1	Mild peribronchiolar cuffing and/or infiltrate in lumina without epithelial degeneration <30% of bronchi/bronchioli
	2	Moderate peribronchiolar cuffing and/or infiltrate in lumina without epithelial degeneration >30% of bronchi/bronchioli
	3	Severe peribronchiolar cuffing and/or infiltrate in lumina with epithelial degeneration and necrosis >30% of bronchi/bronchioli
Blood vessels	0	No changes
	1	Perivascular clear spaces (edema) with mild infiltrates of inflammatory cells >30% of large blood vessels
	2	Clear perivascular cuffing in >30% of blood vessels, with minimal vasculitis
	3	Vasculitis in >30% of blood vessels or extensive perivascular cuffing of more than 5 cell layers
Hemorrhage/hemosiderophages (macrophages with hemosiderin)	0	No hemorrhage/hemosiderophages
	1	Mild focal to multifocal hemorrhage/hemosiderophages
	2	Moderate to severe multifocal hemorrhage/hemosiderophages

**Table 3 pathogens-10-00824-t003:** Scoring system used to grade antigen expression in lungs.

Parameter	Score	Extent of Lesions
**General Scores**		
IHC staining	0	No staining
	1	Focal or multifocal staining (<5 foci) in <10% of the tissue
	2	Multifocal staining in 10–40% of the tissue
	3	Multifocal to coalescing staining in 40–70% of the tissue
	4	Diffuse staining in >70% of the tissue

## Data Availability

Not applicable.

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
