# Peer review of "Predictive Value of Precision-Cut Lung Slices for the Susceptibility of Three Animal Species for SARS-CoV-2 and Validation in a Refined Hamster Model"

_pathogens, 2021, doi:10.3390/pathogens10070824_

Round 1

Reviewer 1 Report

This is an interesting manuscript with information which is useful to the wider scientific community regarding the use of precision-cut lung slices. However, there are major flaws in the data analysis. Throughout the entire manuscript the authors have neglected to carry out any proper statistical analysis of the results, as such the conclusions simply cannot be supported by the qualitative statements in the manuscript as it stands.

Line 22: suscepti-bility – remove hyphen

Line 26: in-oculated – remove hyphen

Line 30: semiquantitative - insert hyphen

Lines 44-49: Needs references

Line 52: abovementioned – insert space

Line 97: ‘sensitive method to monitor discomfort.’ – this is too vague, please add details.

Results section 2.1: This is well written, and I appreciate the level of detail including the Genbank accession number. But this belongs in the methods not the results section.

Figure 1b and c: Why is this presented differently to the viral load data in Figure 6? Please justify or harmonise the presentation of viral load data throughout the manuscript, otherwise this is very confusing. There is absolutely no attempt to carry out statistical analysis, please show some evidence of determining significance.

Figure 2: One representative individual appears to have been shown. The authors need to clarify that data are representative of x number of independent experiments performed on PCLA obtained from x number of animals (i.e. biological replicates). At the very least, they should make some attempt to quantify amount NP in uninfected vs infected PCLS eg NP co-localising with cellular nuclei (DAPI) to give number infected cells per 0.1mm2. Want to see uninfected IF staining (at least in supplemental) and a graph quantifying counts.

Line 159, 461, 492: Throughout the manuscript the authors use infectious doses for the virus (see these lines especially). They do not give any rationale for the selection of these doses – was this determined experimentally? Based on previous literature? Needs clarification.

Line 167-168: Minor point, but it seems strange to compare juvenile and adult animals which have been housed in such different conditions.

Section 2.4.3. Especially referring to Figure 5, the authors jump around (eg Figure 5d is the first figure referred to in this section), please refer to figures in order in the text.

Section 2.4.4. From this point forward the Authors keep referring to ‘animals’ - please specify hamsters, as this vague terminology is unclear and may confuse readers.

Line 229-231: If the authors investigated these tissues, please either show in supplemental data, or state data not shown.

Section 2.4.4. - 2.4.6.: Throughout the manuscript the authors neglect to properly quantify or analyse the results, this is especially obvious in this section. Examples are; “The lungs of all animals  showed extensive staining from DPI 2 until DPI 4 (Figure 5g and l), and mild staining on 250 DPI 7-10, present only in a limited number of animals.” ‘‘Around DPI 10, the RNA levels decreased and on DPI 21 hardly any RNA was detected anymore.” “low levels of antibodies were detected in 2 out of 4 animals as early as DPI  4.” These are incredibly vague statements; the readers need specific values with SEM, range, GMT, OR, CI to get an idea of the actual figures involved.

Lines 407-416: May just be the wording, but it seems like the animals are different SPF statuses. Please reword to clarify. There does not appear to be any ethical oversight for these experiments – please include.

Line 435: “All four elution fractions were pooled’ please explain rationale – are the authors following manufacturers instructions.

Line 477: “precise-cul” Please correct.

Line 478: “Following euthanasia by exsanguination of the pigs, juvenile hamsters and cats” This is extremely concerning ethically, please confirm that an additional method was used (eg anaesthesia).

Table 2a & 2b: These tables belong in the supplemental methods.

Section 4.7 Immunohistochemistry (IHC): Confusingly this is referred to as immunofluorescence staining in the results Figure 2. Please pick one and stick to it.

Section 4.8. The authors appear to have taken tissues at necropsy for RNA isolation and stored them at -70C without any stabilisation, please describe the stabilisation method used on stored tissues (eg RNAlater, trizol etc) to prevent degradation of nucleic acids prior to extraction.

Line 545:  “supplemented with 1% anti/anti.” Please clarify anti/anti.

Lines 547-549: “Lungs were grinded shorter (30 s) since this organ is much softer. The suspensions were cleared by centrifuging for 15 min at 3400xg, 4°C, and then aliquots were either mixed with Trizol-LS (Sigma; Saint 550 Louis, Missouri, USA) in ratio 1:3” Please rewrite this section with better syntax and grammar.

Section 4.10.: This does not appear to be virus isolation; it appears to be viral quantification? Please clarify. This section and section 4.12. are related, but have been organised in a very confusing manner. Please clarify.

Lines 633-637: This is a very confusing sentence, please reword it.

Author Response

Answer to reviewer 1

We thank the reviewer for this critical remark. We added statistical analyses to all graphs in the main body of the text. The description of the analyses methods were added under Materials and Methods (section 4.13) and respective explanations were added to the captions of figures 1, 3, 4, 5, 6.

Line 22: suscepti-bility – remove hyphen

We removed the hyphen.

Line 26: in-oculated – remove hyphen

We removed the hyphen.

Line 30: semiquantitative - insert hyphen

We inserted the hyphen.

Lines 44-49: Needs references

We inserted two references (#5 and #6), which refer to relevant information on the WHO site.

Line 52: abovementioned – insert space

We inserted a space.

Line 97: ‘sensitive method to monitor discomfort.’ – this is too vague, please add details.

We elaborated further this sentence in the revised manuscript.

Results section 2.1: This is well written, and I appreciate the level of detail including the Genbank accession number. But this belongs in the methods not the results section.

We appreciate this comment. However, given the recent appearance of new SARS-CoV-2 variants, as well as the importance which is attributed by the scientific community to details of virus stocks preparation and characterization, we prefer to present this information in the Result section.

Figure 1b and c: Why is this presented differently to the viral load data in Figure 6? Please justify or harmonise the presentation of viral load data throughout the manuscript, otherwise this is very confusing. There is absolutely no attempt to carry out statistical analysis, please show some evidence of determining significance.

We thank the reviewer for pointing this out. To prevent reader’s confusion about the different graph styles throughout the paper, we modified the graphs in figures 4, 5 and 6 to contain lines and individual values, instead of only individual values. In this way also differences (or the lack thereof) between the adult and juvenile hamsters are more clearly seen and the added statistical analysis is easier to interpret.

Furthermore, we added statistical analysis to Fig. 1b and 1c. Of note, in the revised manuscript we excluded the 72h post infection time point because we missed data from the pig experiment and no statistical analysis could be performed on the 72h time point.

Figure 2: One representative individual appears to have been shown. The authors need to clarify that data are representative of x number of independent experiments performed on PCLA obtained from x number of animals (i.e. biological replicates). At the very least, they should make some attempt to quantify amount NP in uninfected vs infected PCLS eg NP co-localising with cellular nuclei (DAPI) to give number infected cells per 0.1mm2. Want to see uninfected IF staining (at least in supplemental) and a graph quantifying counts.

We thank the reviewer for pointing out that the PCLS replicate information was missing. In the revised manuscript we added the missing replicate information to Section 2.2 and figure legend 2.

We furthermore agree with the reviewer that a quantification of NP staining would provide additional information. However, the presented photographs are intended as qualitative illustrations only. Of note, it is technically very difficult to obtain a lot of replicate data. PCLS slices are already thin and processing them into paraffin blocks and subsequently slicing them further is even more challenging. Quantitative data is provided by the PCR analysis. Finally, we considered the suggestion of this reviewer to add images of uninfected cells but decided not to show them as the specificity (signal to noise ratio) in the presented images is very good (at least to our opinion) and a clear distinction can be made between infected and uninfected cells, since both types are present in the images of the cats and the hamsters.  

Line 159, 461, 492: Throughout the manuscript the authors use infectious doses for the virus (see these lines especially). They do not give any rationale for the selection of these doses – was this determined experimentally? Based on previous literature? Needs clarification.

We thank the reviewer for this question. This experiment was performed in the beginning of the SARS-CoV-2 crisis when labs were still establishing there models. We used the highest dose that we could obtain with our undiluted virus stock. To clarify this point, we added a statement in section 4.4 (shown in red below):

All other hamsters (N=36) were exposed to SARS-CoV-2 via intranasal inoculation with 100 µL of undiluted virus stock (dose of 104.5 TCID50), which is the highest virus concentration that could be achieved with our virus stock. Virus was applied under anesthesia.....

Line 167-168: Minor point, but it seems strange to compare juvenile and adult animals which have been housed in such different conditions.

The only difference between the adult and juvenile hamsters was that 8 juvenile hamsters had an activity wheel in their cage, which we considered as a minor difference. We added a clarifying sentence in revised manuscript (line 186-187). 

Section 2.4.3. Especially referring to Figure 5, the authors jump around (eg Figure 5d is the first figure referred to in this section), please refer to figures in order in the text.

This is a valid remark. In the revised manuscript we paid special attention to the order of figures where possible. We realize that the order of figure 5 do not entirely match the text flow. We think, however, that for the reader it will be the easiest to interpret the figure if the images are next to each other, and the graphs summarizing the scores are not situated in between.

Section 2.4.4. From this point forward the Authors keep referring to ‘animals’ - please specify hamsters, as this vague terminology is unclear and may confuse readers.

We changed the wording to “hamsters” in the entire manuscript, where appropriate.

Line 229-231: If the authors investigated these tissues, please either show in supplemental data, or state data not shown.

Since there were no particular findings in these organs, we consider it as not necessary to show photographs of unaffected tissues. So indeed, we added this to the manuscript (line 266).

Section 2.4.4. - 2.4.6.: Throughout the manuscript the authors neglect to properly quantify or analyse the results, this is especially obvious in this section. Examples are; “The lungs of all animals  showed extensive staining from DPI 2 until DPI 4 (Figure 5g and l), and mild staining on DPI 7-10, present only in a limited number of animals.” ‘‘Around DPI 10, the RNA levels decreased and on DPI 21 hardly any RNA was detected anymore.” “low levels of antibodies were detected in 2 out of 4 animals as early as DPI  4.” These are incredibly vague statements; the readers need specific values with SEM, range, GMT, OR, CI to get an idea of the actual figures involved.

To meet the major criticism of this reviewer, statistical analyses were performed on all datasets throughout the manuscript. The outcome of the analyses were subsequently also incorporated into various paragraphs of the results section. The statistical methods used are described under the Material and Methods section.

Lines 407-416: May just be the wording, but it seems like the animals are different SPF statuses. Please reword to clarify. There does not appear to be any ethical oversight for these experiments – please include.

This is correct, hamsters and cats were SPF and pigs were SPF-like from a farm with high health status. There are no “real” SPF pigs available, thus the best option is to use SPF-like pigs. The ethical licenses are mentioned at the end of the manuscript in a separate section (Institutional Review Board Statement).

Line 435: “All four elution fractions were pooled’ please explain rationale – are the authors following manufacturers instructions.

Multiple elutions from a single column followed by a concentration is a frequently used method to increase the yield of RNA/DNA during isolation procedures. We apply this method if we are uncertain if the quantity and/or quality of the RNA/DNA will be sufficient for NGS after a single elution.

Line 477: “precise-cul” Please correct.

Corrected.

Line 478: “Following euthanasia by exsanguination of the pigs, juvenile hamsters and cats” This is extremely concerning ethically, please confirm that an additional method was used (eg anaesthesia).

Thank you for pointing this out. Indeed, animals were deeply anesthesized. We added this missing information to the revised version of the manuscript (lines 547-549).

Table 2a & 2b: These tables belong in the supplemental methods.

We appreciate this remark, however, we consider the pathological scoring scheme as one of the core contributions of this manuscript and therefore prefer to have the tables as part of the main text, so that these tables can be accessed easily by the reader.

Section 4.7 Immunohistochemistry (IHC): Confusingly this is referred to as immunofluorescence staining in the results Figure 2. Please pick one and stick to it.

Thank you for noticing this. In fact, we have performed both IHC (for in vivo organs) as well as IF (for PCLS), but we have not described the methods for immunofluorescence staining. We therefore added additional information to section 4.7 in which we now also describe IF.

Section 4.8. The authors appear to have taken tissues at necropsy for RNA isolation and stored them at -70C without any stabilisation, please describe the stabilisation method used on stored tissues (eg RNAlater, trizol etc) to prevent degradation of nucleic acids prior to extraction.

Indeed, we have taken organ samples upon necropsy for the purpose of RNA isolation. Importantly, in this study, we did not attempt to measure host cell RNA which is degraded rapidly, but viral RNA. RNA viruses escape cellular RNA degradation through various mechanisms (see e.g. doi: 10.1016/j.tig.2011.04.003), and therefore, coronaviral RNA does not need additional stabilization during processing.

Line 545:  “supplemented with 1% anti/anti.” Please clarify anti/anti.

This is antibiotic/antimyctic solution from Gibco. We specified this at all three instances in the methods section (lines 560-561).

Lines 547-549: “Lungs were grinded shorter (30 s) since this organ is much softer. The suspensions were cleared by centrifuging for 15 min at 3400xg, 4°C, and then aliquots were either mixed with Trizol-LS (Sigma; Saint 550 Louis, Missouri, USA) in ratio 1:3” Please rewrite this section with better syntax and grammar.

As suggested, we re-wrote this section to make it clearer to the reader.

Section 4.10.: This does not appear to be virus isolation; it appears to be viral quantification? Please clarify. This section and section 4.12. are related, but have been organised in a very confusing manner. Please clarify.

We modified the title of section 4.10 to “Virus isolation and quantification”

Indeed, both sections 4.10 and 4.11 refer to 4.12, because both assays use the same staining technique for visualization. To improve the text flow, we exchanged the places of sections 4.11 and 4.12.

Lines 633-637: This is a very confusing sentence, please reword it.

Thank you for pointing this out. We now made two sentences from the long and confusing sentence.

Reviewer 2 Report

This is an excellent model development paper and the PCLS technique is a unique addition.  The report is very thorough and well written.  I have only minor comments, as follows:

1) In the methods, please state how the virus was applied to the nares (1 side, or divided over both sides?).

2) Results, section 2.3.  How to the infected cell types identified in the PCLS hamster and cat samples compare with those reported in the literature for in vivo studies?  This could be described in the discussion.  How do the infected cell types for the PCLS hamster specimens compare to the in vivo model published here?

3) Section 2.4.1 Please mention group sizes.  Also, edit lines 163-168 for clarity.  Instead of "returned to levels before infection", recommend "returned to baseline levels" etc.  

4) Section 2.4.2 Please mention group sizes and the days of necropsy.  I realize this is described elsewhere but some group size info would add clarity.

5) Figure 3a:  overall very nice schematic.  Although the n= was somewhat confusing.  Days 2-4, it was  not clear that n=2 per day were being taken.  Also for day 21, should this be n=8?  Additionally, more description for figure 3a in the legend would be helpful describing n=  on which days etc.

6) Section 2.4.5 expand statements where "no virus could be isolated" to state the assay being used and that tissue homogenates were assayed.  There are about 4 such statements in this section that could be clarified as such.

7) Figure 6 legend.  Was VNT defined elsewhere?

8) Discussion line 356:  were temperature decreases statistically significant?

9) Were temperatures taken around the same time each day?  Important to mention.

10) Section 4.4:  What kind of temperature transponders were used.  How were these read?  (and what time of day; range?).

11)  Section 4.8:  What kind of swabs (product info) were used?  Important for replicating the data.

12) Discussion, line 349:  A more thorough list of hamster models would be helpful.  Have others seen this drop in activity?  How does this study compare with regards to the activity measurement and temperature drop?

Author Response

We thank this reviewer for his/her thorough feedback. We agree and appreciate his/her suggestions to improve the manuscript.

This is an excellent model development paper and the PCLS technique is a unique addition.  The report is very thorough and well written.  I have only minor comments, as follows:

1) In the methods, please state how the virus was applied to the nares (1 side, or divided over both sides?).

Thank you for this remark. In revised manuscript we specified this in section 4.4.

2) Results, section 2.3.  How to the infected cell types identified in the PCLS hamster and cat samples compare with those reported in the literature for in vivo studies?  This could be described in the discussion.  How do the infected cell types for the PCLS hamster specimens compare to the in vivo model published here?

Cells were identified based on morphology by certified veterinary pathologists. With respect to the comparison to other in vivo studies, we had addressed this in the discussion already:

“(…)

Next to PCLS from pigs, we also incubated PCLS derived from domestic cats with SARS-CoV-2. The virus efficiently replicated in cat pulmonary tissue, probably even more efficient compared to hamsters, although we have to state that the data is obtained from a single donor due to limited accessibility of cats. The cat PCLS results are in agreement with experimental findings by various groups [21,24-26]. Interestingly, viral antigen was detected in alveoli in cat PCLS, whereas in in vivo infected cats, only acinar glands seem to support virus replication [24]. We speculate that the buffer/agarose gel of the ex-vivo system might have influenced the viral attachment. An-other speculation is that the alveolar walls are sensitive also in vivo, but that the limited viral replication in the large bronchi in vivo [21,24] hampers the virus infection to affect the deeper airways. Finally, PCLS are deprived from immune cell infiltrates, which might modify the dynamics of virus replication in vivo in the cat lungs.

In addition to efficient replication in cat PCLS, SARS-CoV-2 also efficiently replicated in PCLS derived from hamsters. Virus was detected in the alveolar and bronchiolar epithelium, similarly to in vivo infected hamsters, where virus was predominantly found in the same type of cells as observed by us and others [27-29].

(…)“

3) Section 2.4.1 Please mention group sizes.  Also, edit lines 163-168 for clarity.  Instead of "returned to levels before infection", recommend "returned to baseline levels" etc.  

We modified section 2.4.1 as suggested.

4) Section 2.4.2 Please mention group sizes and the days of necropsy.  I realize this is described elsewhere but some group size info would add clarity.

We modified section 2.4.2 as suggested

5) Figure 3a:  overall very nice schematic.  Although the n= was somewhat confusing.  Days 2-4, it was  not clear that n=2 per day were being taken.  Also for day 21, should this be n=8?  Additionally, more description for figure 3a in the legend would be helpful describing n=  on which days etc.

We agree that this schematic could be unclear. We specified the n=2 necropsy days on DPI 2, 3, 4 and n=8 on DPI 21 in the schematic, and expanded the figure legend.  

6) Section 2.4.5 expand statements where "no virus could be isolated" to state the assay being used and that tissue homogenates were assayed.  There are about 4 such statements in this section that could be clarified as such.

Thank you, we expanded our statement for the entire section 2.4.5.  

7) Figure 6 legend.  Was VNT defined elsewhere?

This was only mentioned in the methods section. To make it clear, we replaced this one instance by “virus neutralization tests”

8) Discussion line 356:  were temperature decreases statistically significant?

Statistical analysis was added to Figure 3 and respective data interpretation in the Result section 2.4.1

9) Were temperatures taken around the same time each day?  Important to mention.

Thank you very much for this remark. We clearly missed to clarify the temperature measurement interval. We therefore modified the methods section 4.4.

10) Section 4.4:  What kind of temperature transponders were used.  How were these read?  (and what time of day; range?).

Please see answer to your question 9 above.

11)  Section 4.8:  What kind of swabs (product info) were used?  Important for replicating the data.

We added this information to the methods section 4.8.

12) Discussion, line 349:  A more thorough list of hamster models would be helpful.  Have others seen this drop in activity?  How does this study compare with regards to the activity measurement and temperature drop?

The measurement of activity in hamsters post SARS-CoV-2 infection is not described yet to our knowledge. Also, most studies do not measure the body temperature, as initial studies have shown that hamsters do not develop fever post infection with SARS-CoV-2. A temperature drop has not been described yet.

Reviewer 3 Report

In Pathogens Manuscript ID # 1234020, Gerhards et al. tested if precision-cut lung slices (PCLS) from hamsters, pigs and cats were supportive of SARS-CoV-2 infection.  They also compared their ex vivo results from the PCLS experiments with in vivo infections using juvenile and aged hamsters.  They found that PCLS from hamsters and cats, but not pigs, support viral replication.  Immunohistochemistry identified infected cells.  No clear differences were observed between juvenile and aged hamsters.  The manuscript is well-written, it’s clear and concise.  The experimental procedures are clearly described.  The results and conclusions are supported by the data.  The manuscript offers additional information regarding the continued development of the hamster model for SARS-CoV-2.  This model is providing important information about potential treatments for COVID-19 and could be crucial in helping combat a future CoV pandemic.  The authors need to address the 1 minor comment/suggestion that I list below.

Comments/Suggestions:

  1. In figure 1b and 1c, the authors compare viral replication in PCLS from cats, hamsters and pigs. There appears to be increased viral replication in cat PCLS compared with hamsters, and in lines 119-120, the authors state that the virus replicated more rapidly in cats. Did the authors compare peak viral titers, the slope of the curve to peak, or area under the curve to determine whether the observed difference is statistically significant?

Author Response

We thank this reviewer for his/her positive feedback and addressed his/her question below.

Comments/Suggestions:

1. In figure 1b and 1c, the authors compare viral replication in PCLS from cats, hamsters and pigs. There appears to be increased viral replication in cat PCLS compared with hamsters, and in lines 119-120, the authors state that the virus replicated more rapidly in cats. Did the authors compare peak viral titers, the slope of the curve to peak, or area under the curve to determine whether the observed difference is statistically significant?

We thank the reviewer for this important comment, which was also a comment of the other reviewers. As stated before, we now performed statistical analysis on all data sets including the data presented in Figure 1b and c. We expanded the results section with the statistical analysis and expanded figure legends, Furthermore, details of the statistical analysis methods are provided in the Material and Methods section.

Round 2

Reviewer 1 Report

The authors have addressed the issues raised by the reviewers to my satisfaction.